# Lesion-Function Analysis from Multimodal Imaging and Normative Brain Atlases for Prediction of Cognitive Deficits in Glioma Patients

**DOI:** 10.3390/cancers13102373

**Published:** 2021-05-14

**Authors:** Martin Kocher, Christiane Jockwitz, Philipp Lohmann, Gabriele Stoffels, Christian Filss, Felix M. Mottaghy, Maximilian I. Ruge, Carolin Weiss Lucas, Roland Goldbrunner, Nadim J. Shah, Gereon R. Fink, Norbert Galldiks, Karl-Josef Langen, Svenja Caspers

**Affiliations:** 1Institute of Neuroscience and Medicine (INM-4), Research Center Juelich, 52428 Juelich, Germany; p.lohmann@fz-juelich.de (P.L.); g.stoffels@fz-juelich.de (G.S.); c.filss@fz-juelich.de (C.F.); n.j.shah@fz-juelich.de (N.J.S.); k.j.langen@fz-juelich.de (K.-J.L.); 2Department of Stereotaxy and Functional Neurosurgery, Center for Neurosurgery, Faculty of Medicine and University Hospital Cologne, 50937 Cologne, Germany; maximilian.ruge@uk-koeln.de; 3Center of Integrated Oncology (CIO), Universities of Aachen, Bonn, Cologne and Duesseldorf, 50937 Cologne, Germany; carolin.weiss-lucas@uk-koeln.de (C.W.L.); roland.goldbrunner@uk-koeln.de (R.G.); gereon.fink@uk-koeln.de (G.R.F.); norbert.galldiks@uk-koeln.de (N.G.); 4Institute of Neuroscience and Medicine (INM-1), Research Center Juelich, 52428 Juelich, Germany; c.jockwitz@fz-juelich.de (C.J.); s.caspers@fz-juelich.de (S.C.); 5Department of Nuclear Medicine, University Hospital Aachen, RWTH Aachen University, 52074 Aachen, Germany; fmottaghy@ukaachen.de; 6Department of Radiology and Nuclear Medicine, Maastricht University Medical Center, 6229 HX Maastricht, The Netherlands; 7Department of Neurosurgery, Center for Neurosurgery, Faculty of Medicine and University Hospital Cologne, 50937 Cologne, Germany; 8Department of Neurology, University Hospital Aachen, RWTH Aachen University, 52074 Aachen, Germany; 9Juelich-Aachen Research Alliance (JARA)–Section JARA-Brain, 52428 Juelich, Germany; 10Institute of Neuroscience and Medicine (INM-3), Research Center Juelich, 52428 Juelich, Germany; 11Department of Neurology, Faculty of Medicine and University Hospital Cologne, University of Cologne, 50937 Cologne, Germany; 12Institute for Anatomy I, Medical Faculty & University Hospital Düsseldorf, Heinrich Heine University Duesseldorf, 40225 Duesseldorf, Germany

**Keywords:** glioma, brain networks, positron emission tomography, radiotherapy, cognitive testing

## Abstract

**Simple Summary:**

This prospective cross-sectional study utilized standard structural MR imaging and amino acid PET in conjunction with brain atlases of gray matter functional regions and white matter tracts, and elastic registration techniques to estimate the influence of the type and location of treatment-related brain damage or recurrent tumors on cognitive functioning in a group of well-doing WHO Grade III/IV glioma patients at follow-up after treatment. The negative impact of T2/FLAIR hyperintensities, supposed to be mainly caused by radiotherapy, on cognitive performance far exceeded that of surgical brain defects or recurrent tumors. The affection of functional nodes and fiber tracts of the left hemisphere and especially of the left temporal lobe by T2/FLAIR hyperintensities was highly correlated with verbal episodic memory dysfunction. These observations imply that radiotherapy for gliomas of the left hemisphere should be individually tailored by means of publicly available brain atlases and registration techniques.

**Abstract:**

Cognitive deficits are common in glioma patients following multimodality therapy, but the relative impact of different types and locations of treatment-related brain damage and recurrent tumors on cognition is not well understood. In 121 WHO Grade III/IV glioma patients, structural MRI, *O*-(2-[18F]fluoroethyl)-L-tyrosine FET-PET, and neuropsychological testing were performed at a median interval of 14 months (range, 1–214 months) after therapy initiation. Resection cavities, T1-enhancing lesions, T2/FLAIR hyperintensities, and FET-PET positive tumor sites were semi-automatically segmented and elastically registered to a normative, resting state (RS) fMRI-based functional cortical network atlas and to the JHU atlas of white matter (WM) tracts, and their influence on cognitive test scores relative to a cohort of matched healthy subjects was assessed. T2/FLAIR hyperintensities presumably caused by radiation therapy covered more extensive brain areas than the other lesion types and significantly impaired cognitive performance in many domains when affecting left-hemispheric RS-nodes and WM-tracts as opposed to brain tissue damage caused by resection or recurrent tumors. Verbal episodic memory proved to be especially vulnerable to T2/FLAIR abnormalities affecting the nodes and tracts of the left temporal lobe. In order to improve radiotherapy planning, publicly available brain atlases, in conjunction with elastic registration techniques, should be used, similar to neuronavigation in neurosurgery.

## 1. Introduction

Patients suffering from WHO Grade III/IV glioma typically undergo an extended sequence of therapeutic interventions, including repeated tumor resection, irradiation, re-irradiation, and multiple courses of chemotherapy or targeted molecular therapy [1,2,3]. While tumor resection carries the risk of immediate damage to functional gray matter and the connecting tracts, radiotherapy is assumed to induce delayed damage, predominantly to the irradiated white matter [4]; furthermore, chemotherapy may cause diffuse damage in widespread regions of the brain [5]. There is a general consensus that short-term impairment of neurological functioning can be avoided in most cases by sparing eloquent brain areas from resection [6,7,8], but the long-term cognitive outcome is far less predictable from the extent of resection [9,10,11,12], the location and dose of radiotherapy [13,14,15], or the type and intensity of chemotherapy [5]. Importantly, recurrent tumor growth itself may also impact cognition [13].

According to current brain function theories, performance in higher cognitive domains depends on the specialized function of cortical regions and their interaction in multiple networks [16]. Recently, a parcellation of the cortex into functionally connected nodes belonging to specified networks of known cognitive function was derived from resting-state fMRI data [17,18]. Moreover, a widely recognized atlas of the central white matter tracts was developed by the Johns Hopkins University [19].

As the extent and localization of the different types of brain damage (resection cavities, radiation-induced white matter damage, tumor infiltration) can be determined with the aid of automated image segmentation methods [20], we hypothesize here that the normative brain atlases characterizing the functional nodes and structural connectivity of the brain can be used in conjunction with multimodal brain imaging and segmentation techniques to predict cognitive deficits resulting from treatment or recurrent tumor growth, and will allow the identification of brain regions that are particularly vulnerable in terms of cognition [21,22].

## 2. Materials and Methods

### 2.1. Study Population

This prospective, cross-sectional study recruited patients suffering from WHO Grade III or IV glioma referred for hybrid MR/PET imaging for therapy monitoring or suspicion of a recurrent tumor after initiation of therapy. Patients were screened by phone calls and reviewed at the date of imaging. The inclusion criteria were: good general condition (ECOG performance score, 0–1 [23]), no major depression, no seizures, and fluent in the German language. The local ethics committee approved the protocol and all patients (and healthy controls; see below) provided informed written consent following the Declaration of Helsinki. From 6 February 2018 to 2 September 2020 (31 months), 121 patients were enrolled. As shown in Appendix A, the median interval between therapy initiation and imaging was 14 months (mean, 30 months; range, 1–214 months), and 105 patients (87%) had completed tri-modality primary treatment, including biopsy/resection, adjuvant radiotherapy, and chemotherapy. Moreover, by this time, most patients had received 1 series (100 patients, 83%) or 2 series (14 patients, 11%) of local radiotherapy (59–62 Gy in 92%) during their disease course, with a median interval of 13 months (mean, 32 months; range, 2–213 months) between the start of irradiation and imaging. All patients except 1 were right-handed.

As preoperative cognitive data were not available for the patients, a group of healthy subjects from a population-based cohort study that investigated environmental and genetic influences on the inter-individual variability of brain structure, function, and connectivity in the aging brain (1000 BRAINS study, [24]), which matched the patient population in terms of gender, age, and education, was used for comparison.

### 2.2. FET-PET and MR Imaging

As described in detail before [25], *O*-(2-[18F]fluoroethyl)-l-tyrosine (FET) PET images were obtained using a high-resolution 3T hybrid PET/MR scanner (Siemens Tim-TRIO/BrainPET, Siemens Medical Systems, Erlangen) equipped with a PET insert. The presence or absence of active tumor sites was assessed by a nuclear medicine physician (K.J.L.) [26]. MR data were acquired simultaneously with the PET data on the high-resolution 3T hybrid PET/MR scanner equipped with a birdcage-like quadrature transmitter head coil and an 8-channel receiver coil. Structural MR imaging included a 3D T1-weighted magnetization-prepared rapid acquisition gradient-echo (MPRAGE) anatomical scan (176 sagittal slices, TR = 2250 ms, TE = 3.03 ms, FoV = 256 × 256 mm^2^, flip angle = 9°, voxel size = 1 × 1 × 1 mm^3^), a contrast-enhanced T1-weighted image (T1-CE) obtained from a second MPRAGE scan following the injection of gadolinium (0.2 mmol/kg, Dotarem, Guerbet GmbH, Sulzbach, Germany) or high-resolution T1-weighted, contrast-enhanced MR scans available from the referring institution. In addition, T2-weighted (T2-SPACE, 176 slices, TR = 3.2 s, TE = 417 ms, FoV = 256 × 256 mm^2^, voxel size = 1 × 1 × 1 mm^3^) and T2-weighted fluid-attenuated structural images (T2-FLAIR, 25 slices, TR = 9000 ms, TE = 3.86 ms, FoV = 220 × 220 mm^2^, flip angle = 150°, voxel size = 0.9 × 0.9 × 4 mm^3^) were acquired.

### 2.3. MRI and FET-PET Image Segmentation

Lesion masks were generated for resection cavities, T1-CE enhancing lesions, T2/FLAIR hyperintensities, and FET-PET-positive tumor sites. Before segmentation, all MR and PET images were registered to the T1 image using FSL-FLIRT (FSL toolbox, FMRIB Software Library vs. 5.0, Oxford, UK, http://www.fmrib.ox.ac.uk/fsl (accessed on 1 February 2018)) [27]. Resection cavities were manually contoured by a radiation oncologist (M.K.) using itk-SNAP (http://www.itksnap.org, vs. 3.8.0, Universities of Pennsylvania and Utah, Philadelphia, USA (accessed on 1 February 2019)). The T1-CE-enhancing lesions and T2/FLAIR-related tissue (hyperintensities) were automatically segmented using the deep-learning-based software HD-GLIO-AUTO (https://github.com/NeuroAI-HD/HD-GLIO-AUTO, Department of Neuro-radiology at the Heidelberg University Hospital, and Division of Medical Image Computing at the German Cancer Research Center (DKFZ) Heidelberg, Germany (accessed on 1 February 2020)) after the application of brain extraction by HD-BET (https://github.com/MIC-DKFZ/HD-BET, developed by the same group (accessed on 1 February 2020)) [20,28]. Automatic FET-PET segmentation was based on an empirically derived threshold for a tumor-to-brain ratio (TBR) of 1.6, which is highly predictive of glioma tissue [29] and was implemented by an FSL script (see Appendix B). All automatic MR and FET-PET segments were visually inspected and manually corrected if necessary using itk-SNAP.

### 2.4. Determination of Lesion-Specific Damage to Functional Cortical Regions and White Matter Tracts

For function–lesion mapping, the functional RS-fMRI-based cortical parcellation of Schaefer [17] (2 × 50 parcels in the lowest resolution) belonging to 7 resting-state networks (visual, somato-motor, dorsal attention, ventral attention, limbic, fronto-parietal control, and default mode) and the white matter tracts (2 × 24 tracts) from the Stereotaxic White Matter Atlas of the Johns Hopkins University (JHU) [19] were used. All images were spatially normalized by elastic registration to the MNI-152 brain template by means of the SPM12 toolbox (Statistical Parametric Mapping Toolbox, MatLabR R2017b, MathWorks, Natick, MA, USA, www.fil.ion.ucl.ac.uk/spm/software/spm12 (accessed on 1 February 2018)) [30,31] and, finally, the partial overlapping volume of the segments with the nodes or tracts was computed (Figure 1A).

The relationship of socio-economic (age, gender, education) factors, clinical factors, or total volume of the different lesion types with cognitive outcome was assessed by Spearman’s rank correlation (two-sided) in the case of continuous or ordinal variables, the Mann–Whitney U-test (two-sided) in case of dichotomous variables, and the Kruskal–Wallis test (two-sided) for categorial variables with more than two groups. Apart from the abovementioned correlation measures that served for the identification of possible predictive variables, multiple logistic regression models were trained to predict the probability of clinically relevant cognitive impairment. Classification performance was analyzed by contingency tables and the receiver–operator characteristics (ROC) method.

### 2.5. Cognitive Assessment

Cognitive functioning was assessed at the date of imaging using a comprehensive subset of a test battery from the 1000 BRAINS study [24] that could be completed within 25–30 min and was designed to assess the main cognitive domains in a reasonable time that was tolerable for the brain tumor patients (Table 1. The 1000 BRAINS study, a population-based cohort study that included over 1300 older subjects, investigated environmental and genetic influences on the inter-individual variability of brain structure, function, and connectivity in the aging brain. From these, a group of 121 healthy subjects was selected who matched the patient population in terms of gender, age, and education by propensity score matching as implemented in the software R [32,33]. Clinically relevant cognitive impairment was assumed if the scores of the patients ranged 1.5 standard deviations below the mean of the healthy subjects. Moreover, educational status was determined according to the 1997 ISCED scoring system. (http://uis.unesco.org/sites/default/files/documents/international-standard-classification-of-education-1997-en_0.pdf, (accessed on 1 February 2018))

### 2.6. Statistical Analysis and Prediction of Cognitive Impairment

SPSS (IBM SPSS Statistics version 27, IBM Corporation, Armonk, NY, USA) and the Python software package scipy.stats (version 1.5.4, https://scipy.org (accessed on 1 February 2020)) were used for all analyses. Comparisons of cognitive test results between patients and healthy subjects were made using the Mann–Whitney U-test (2-sided). Correlation coefficients between cognitive test scores and the volumetric overlap between lesions and functional nodes or white matter tracts were computed using a 2-sided Kendall tau-b rank correlation that automatically corrects for ties and is less sensitive to outliers, and subjected to Bonferroni correction, which took the total number of tests into account [39].

## 3. Results

### 3.1. Cognitive Test Scores in WHO Grade III/IV Glioma Patients

Age (51.7 + 11.5, 51.6 + 11.6 years, two-sided *t*-test *p* = 0.96), gender (male/female 75/46, 73/48, two-sided Chi-square test *p* = 0.90), and educational status (ISCED-score, 7.4 + 1.7, 7.1 + 2.1, 2-sided Mann–Whitney U-test *p* = 0.41) were evenly distributed in the control and patient groups. As shown in Table 1, scores for most cognitive tests (9/10) were significantly lower and more variable in the patients, and affected 10–47% of the patients by a clinically relevant deficit. Age and educational status were significantly rank-correlated with cognitive functioning in most domains (Appendix A, Figure 2). Neither gender, number of radiotherapy series received, number of chemotherapy courses, total number of oncologic interventions, use of anticonvulsants, nor the time interval between therapy initiation and imaging significantly impacted cognitive performance. A weak influence was observed for glioma grading and molecular subtype (anaplastic astrocytoma/glioblastoma, mutated IDH vs. wild-type IDH vs. anaplastic oligodendroglioma) on the TMT-A scores; see Appendix A.

### 3.2. Impact of Total Lesion Volumes and Node or Tract Affection on Cognitive Function

The volume of the T2/FLAIR hyperintensities (median, 53.4; mean, 70.1; range, 3.4–252.9 mL) was significantly (*p* < 0.001, two-sided Kruskal-Wallis test) larger than that of the resection cavities or recurrent tumor volumes (Appendix A). The T2/FLAIR hyperintensities were mainly located in the white matter, but frequently extended up to the white–gray matter border and thus also affected the cortical functional nodes (Figure 1). As shown in Appendix A, the total volume of the T2/FLAIR hyperintensities was significantly correlated with the scores of 8/10 tests, where it prolonged the time needed for completion of the trail-making tests and reduced the scores for the other tests. The same pattern, although with much lesser impact in terms of the strength of correlation and number of test scores affected, was observed for the total volumes of tumor tissue as detected by T1-contrast enhanced MR images or FET-PET. Unexpectedly, in the opposite direction, the patients with larger resection cavities performed better in 3/10 tests than those with smaller cavities.

Typically, any single node or tract was affected in only a small number of patients. In those unaffected, the distribution of cognitive scores resembled that of the matched healthy controls (Figure 2). Heat-maps for the *p*-values of the Kendall tau-b correlations for all combinations of cognitive test scores and volumetric overlaps are provided in Figure 3 and Figure 4. T2/FLAIR lesions most significantly correlated with cognition compared with the other lesion types. After Bonferroni correction, all significant correlations of test scores were related to the T2/FLAIR signals (19 nodes and 7 white matter tracts). They were uniformly located in the left hemisphere and involved three cognitive tests in two domains: verbal episodic memory (immediate recall, eight nodes and four tracts; delayed recall, nine nodes and three tracts), language/executive functioning (number transcoding, two nodes). The nodes and tracts involved in the T2/FLAIR-related impairment of verbal episodic memory (immediate recall) are depicted in Figure 1B and were all located in or close to the left temporal lobe. As T2/FLAIR changes may reflect different pathologies such as edema, gliosis, and low-grade tumor growth, the analysis was repeated in the subgroup of 63 patients who were free from tumor growth as diagnosed by the FET-PET and where the T2/FLAIR changes were therefore most probably caused only by radiation therapy. The association pattern between cognitive deficits and functional node affection resembled that of all patients (Figure 3 and Figure 4, second row).

A different pattern was observed for visual memory impairment, which correlated mainly with right hemispheric T2/FLAIR changes affecting selected nodes of the visual, dorsal attention, ventral attention, frontal control and default mode networks. The scores for the trail-making Tests A and B, reflecting several cognitive abilities (e.g., attention, visual search and scanning, sequencing and shifting, psychomotor speed, and flexibility [40]), varied more substantially in the tumor patients and showed a weak dependency on several left-hemispheric functional nodes.

### 3.3. Prediction of Cognitive Impairment

The partial volumetric overlaps of the abovementioned 26 nodes and tracts with T2/FLAIR changes were highly correlated (Pearson correlation, *p* < 0.001), preventing them for usage as independent predictive variables for cognitive outcomes. Therefore, logistic regression models for clinically relevant cognitive impairment in the abovementioned domains were computed by including the variables age, educational status, and total lesion volumes alone or combined with the partial volumetric affection of one representative node or tract by T2/FLAIR hyperintensities. As shown in Table 1, 17–32% of the patients had clinically relevant lowered scores. By including a single representative node or tract affection, the models for verbal episodic memory and number transcoding improved markedly due to increases in sensitivity by 9–27% (Table 2).

## 4. Discussion

### 4.1. Main Findings

In the present analysis performed in WHO Grade III/IV glioma patients following multimodality therapy, T2/FLAIR abnormalities rather than brain tissue defects due to neurosurgical resection or recurrent tumors most severely correlated with cognitive outcomes. Most of the atlas-derived functional cortical nodes and white matter tracts that significantly influenced cognitive performance were left-hemispheric. Verbal episodic memory and other language-related functions proved to be especially vulnerable to T2/FLAIR hyperintensities affecting the left temporal lobe’s nodes and tracts. The same dependency pattern was recognized in a patient subgroup where the T2/FLAIR abnormalities were observed in the absence of tumor recurrence and were thus probably mainly related to radiotherapy.

### 4.2. Treatment-Related Tissue Damage and Recurrent Tumors as Causes for Cognitive Deficits

The clinical course of patients suffering from newly diagnosed WHO Grade III/IV gliomas includes the initial presentation with neurological symptoms and cognitive deficits due to both the tumor, as well as peritumoral edema and mass effects; the initiation of multimodal therapy including biopsy or resection, local radiotherapy, and simultaneous and/or adjuvant chemotherapy; and early recovery from the preoperative deficits [9]. During the disease’s further course, additional deficits may develop due to a recurrent tumor or treatment-related side-effects. The presence of cognitive deficits in treated WHO Grade III/IV patients was clearly demonstrated in our analysis, as the patients performed significantly lower in the majority of domains compared with a matched group of healthy subjects.

Neurosurgical tumor resection carries a high risk of unintended brain damage. However, nowadays, preoperative mapping techniques are widely used to avoid injury to eloquent functional cortical regions and the principal white matter tracts that are involved in language processing and movement control [6,41,42,43]. Moreover, functional reorganization may start early after surgery. This plasticity is probably why, as in the present investigation, the few reports trying to predict the long-term cognitive outcome from the resection cavity’s location and size have failed to detect critical brain regions [9,10,11].

The situation is different for radiotherapy in several aspects. Here, the long-term sequelae are caused by slowly progressing, vessel-related damage to the white matter, resulting in demyelination, focal necrosis, and gliosis [4]. These changes probably counteract functional reorganization and tend to worsen deficits over time. Moreover, radiation target volumes are usually much more extensive than resection cavities and comprise larger volumes of normal brain tissue, where microscopic tumor infiltration is only assumed. Radiation-induced white matter changes typically present as areas with increased signal intensity in T2-weighted and FLAIR images [44,45,46]. These changes most significantly impacted cognitive functioning in several domains in the present study. As white matter hyperintensities may also have been caused by peritumoral edema at recurrent tumor sites, the analysis was repeated in the smaller group of patients free from tumor recurrence as determined by FET-PET criteria, where a similar pattern of dependencies could be revealed.

Concerning tumor recurrence as a possible cause of cognitive impairments, our patient cohort represents a subpopulation of patients at follow-up after treatment for WHO Grade III/IV glioma. Most patients were in good condition, free of severe neurological deficits, performed well in daily life, and participated in a follow-up program that included repeated combined PET/MR imaging. Therefore, tumor recurrences were detected early, i.e., often before causing new symptoms or deficits, and were only weakly related to cognitive test scores.

### 4.3. Impact of Lesion Location on Cognitive Deficits

The results presented here show that (1) left hemispheric T2/FLAIR lesions cause significantly more cognitive deficits than those of the right hemisphere, (2) these deficits mainly concern verbal episodic memory and other language functions, and (3) T2/FLAIR hyperintensities affecting nodes and tracts located in the left temporal lobe have the most significant impact on performance in these domains. Several other studies conducted in the pre- or early postoperative phase in glioma patients have revealed similar results concerning the location of critical regions. In a mixed collective of low- and high-grade diffuse gliomas evaluated preoperatively by tumor localization maps, a broad range of cognitive domains were affected. Most deficits were related to tumors invading cortical regions and subcortical fiber tracts of the left hemisphere [47]. In a study on high-grade gliomas, deficits in language and verbal memory tasks were mainly caused by left temporal and insular tumors and resolved with a decreasing volume of the surrounding edema [9]. In another voxel-based lesion analysis for verbal-auditory short-term memory and noun/verb processing in patients with resection of left- or right-sided gliomas, operative lesioning of the left posterior-superior temporal cortex, the left supra-marginal gyrus of the inferior parietal lobule, and the inferior longitudinal fasciculus was found to be critical. In a more recent report on WHO Grade II and III tumors treated by surgery and radiotherapy and evaluated 6 months after surgery, most of the cognitive functions involving speech and memory were associated with lesions in the left temporal lobe and supra-marginal gyrus in left-sided tumors. In contrast, surgical lesions of the right frontal cortex caused lower performance in concept shifting and other tasks [48].

These data are based on anatomical rather than functional cortical parcellations and they rarely take the differential effect of the type of lesion involved into account. Moreover, according to modern views of brain function, performance in higher cognitive function depends on the activity and integrity of networks of functional cortical subunits rather than on the performance of single cortical areas [16,49,50,51]. Therefore, we expected to find a distributed pattern of dependencies of cognitive functions on the integrity of certain nodes and tracts. Indeed, the performance in verbal episodic memory and number transcoding was especially significantly influenced by a large number of nodes belonging to the left-sided parts of the visual, somatomotor, dorsal/ventral attention, fronto-parietal, or default-mode networks. However, all nodes and tracts with the most significant impact on these cognitive functions were located in or close to the left temporal lobe, including the fornix and the superior longitudinal fasciculus, which are implicated in working memory and language articulation.

However, the kind of detailed analysis applied here is hampered by the fact that the T2/FLAIR tended to extend over large regions covering many neighboring nodes and tracts to a similar extent. This leads to a highly correlated affection of these nodes in the overall dataset, which renders a meaningful higher-dimensional analysis such as multiple regression or machine learning methods challenging to perform. An individual assessment of structural and functional connectivity during follow-up would be needed to unravel the effect of lesion patterns on cognitive outcome with the necessary attention to detail.

### 4.4. Limitations of the Study

The present work has several limitations. Most patients were in good general condition (ECOG 0–1) and more than 40% were less than 50 years old, which differed from the typical population of WHO Grade III/IV glioma patients at follow-up. Another limitation is this study’s cross-sectional rather than longitudinal character. The patients included had a large variability in age, glioma type, time since therapy initiation, therapy intensity, and anticonvulsive medication, but had in common that they were in good clinical condition. Thus, patients with a more severe deterioration of general performance were not considered. However, cognitive malfunction is probably more debilitating in the physically well-doing patient group that was investigated here, where, apart from the described influence of lesion type and location, none of the other factors proved to have a major impact on cognitive outcome. As in clinical practice, the true nature of T2/FLAIR enhancing lesions remained somewhat indeterminate, but at least the larger lesions in patients free from recurrent tumors were most probably caused by radiotherapy. Moreover, the true volume of the resected brain tissue may have been underestimated due to postoperative processes including brain shift.

## 5. Conclusions

T2/FLAIR changes covering larger brain areas may lead to significant impairments of cognitive functioning, especially when involving the left-sided temporal lobe. In contrast, tumor recurrence is only a minor cause of cognitive deterioration if detected early by close follow-up with amino-acid PET and MRI. These results were obtained by application of publicly available atlases of functional and structural brain organization and image segmentation and elastic registration methods applied to imaging modalities in clinical use. As T2/FLAIR changes may be caused by late radiation side-effects, individually tailored radiation treatment planning based on PET-imaging and advanced MR techniques [52] should be applied to avoid unnecessary irradiation of the vulnerable regions. However, where these imaging techniques are not available, applying standard MR imaging together with brain atlases may be of significant value and could provide a kind of neuronavigation for the radiotherapist. Regarding current concepts in radiation oncology, bilateral hippocampal sparing radiotherapy [53,54,55] should be supplemented by increased awareness of sparing the left-sided temporal lobe in both patients with glioma and brain metastases.

## Figures and Tables

**Figure 1 cancers-13-02373-f001:**
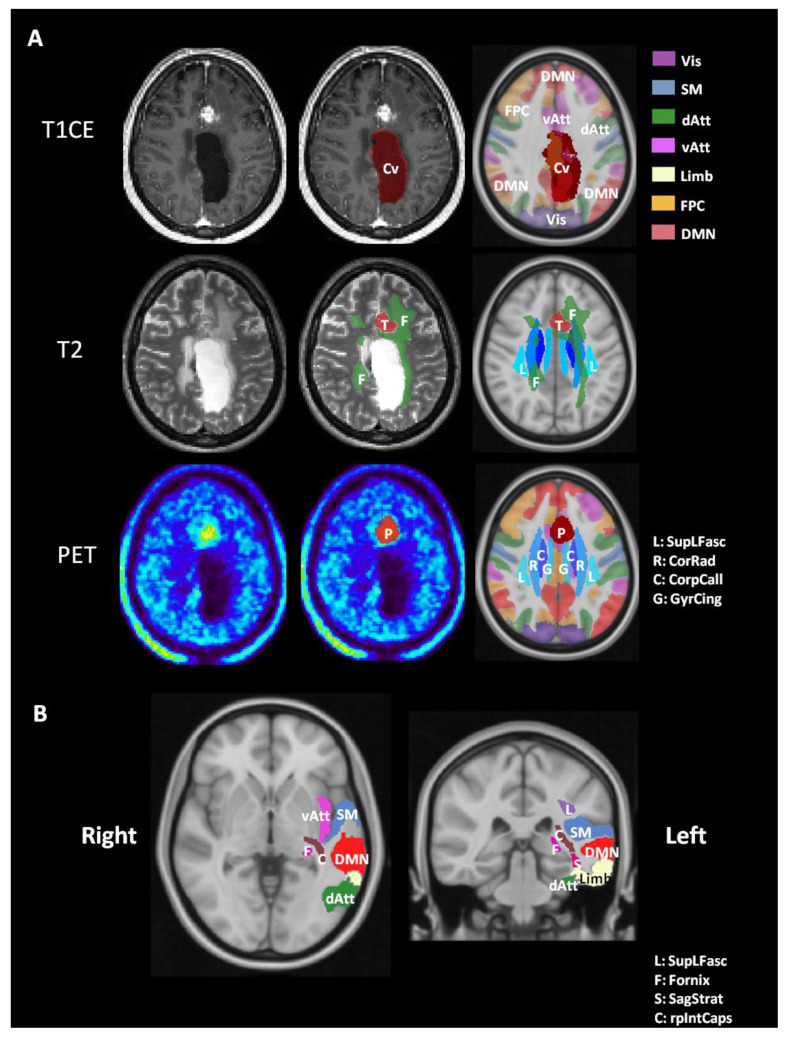
(**A**): Image processing and analysis. Contrast enhancing tumors (T), T2/FLAIR hyperintensities (F), and FET-PET positive tumors (P) were segmented semi-automatically; resection cavities (Cv) were manually outlined. After elastic registration to the MNI-152 space, the partial volumetric overlap of all lesions with 2 × 50 functional resting-state atlas regions (organized into seven networks) and 2 × 24 atlas-based white matter tracts was computed. (**B**): Atlas-based functional cortical nodes and white matter tracts impacting episodic memory (immediate recall) when affected by T2/FLAIR hyperintensities in *n* = 121 WHO Grade III/IV glioma patients. The functional nodes belong to different resting-state networks. T1CE: T1-weighted MR image with contrast enhancement, T2: T2-weighted SPACE MR image, PET: *O*-(2-[18F]fluoroethyl)-L-tyrosine positron-emission tomography; Vis: visual network, SM: somato-motor network, dAtt: dorsal attention network, vAtt: ventral attention network, Limb: limbic network, FPC: fronto-parietal control network, DMN: default mode network, SupLFasc: superior longitudinal fascicle, CorRad: corona radiata, CorpCall: corpus callosum, GyrCing: cingulate gyrus, SagStrat: sagittal stratum, rpIntCaps: retrolenticular part of the internal capsule.

**Figure 2 cancers-13-02373-f002:**
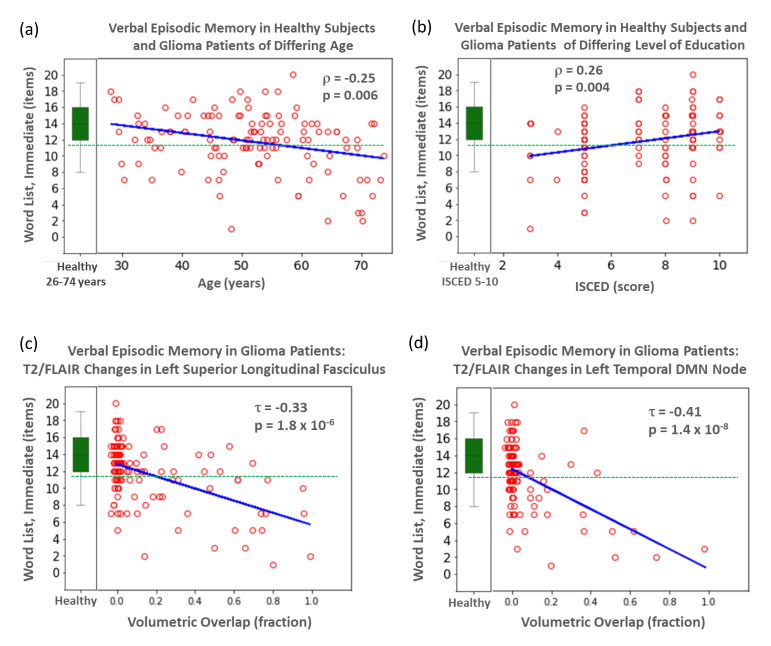
Correlation analysis for test scores of verbal episodic memory in WHO Grade III/IV glioma patients (*n* = 121, red circles) with respect to age (**a**), education level (**b**), and the affection of selected atlas-derived tracts (left superior longitudinal fasciculus, (**c**) or resting-state functional nodes (left temporal DMN node, (**d**). τ: Kendall-tau-b correlation coefficient. ρ: Spearman’s rank correlation coefficient. A small jitter was added to the volumetric overlap in order to prevent overplotting. In addition, the calculated line for linear regression is depicted in blue. For reference, the distribution of scores in healthy subjects is shown as a boxplot (green). Patients were assumed to be affected by clinically relevant cognitive deficits if they fell below a threshold (defined as 1.5 standard deviations below the mean) based on the healthy subjects (green dashed line).

**Figure 3 cancers-13-02373-f003:**
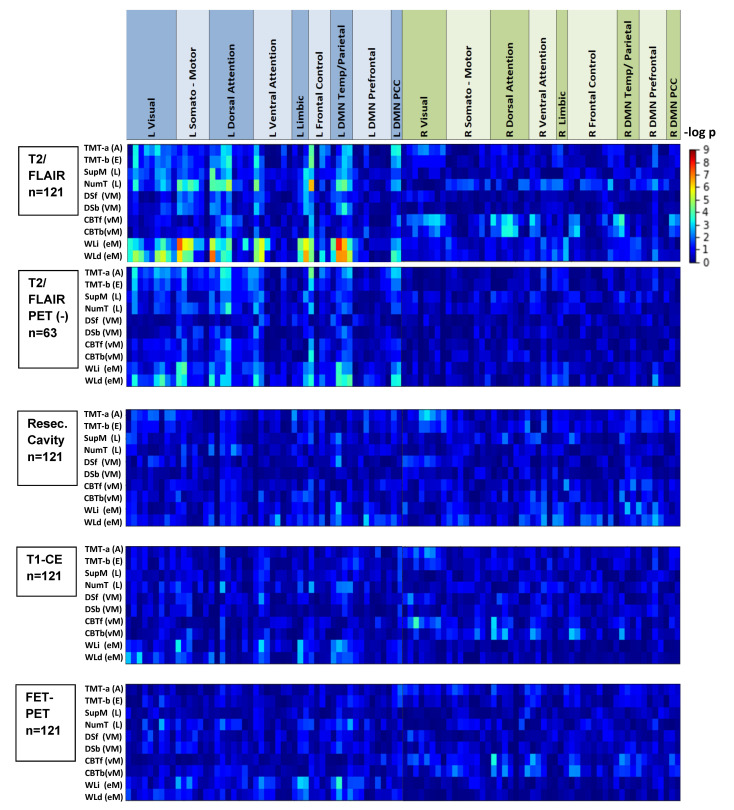
Heatmap of the correlation analysis between the volumetric overlap of four types of lesions with the functional cortical areas of an atlas-based parcellation [17] and the scores of 10 cognitive tests in *n* = 121 WHO Grade III/IV glioma patients. The functional nodes are ordered by location in the left or right hemisphere and the essential resting-state networks. *p*-values are given for the Kendall-tau-b rank correlations. T2/FLAIR: lesions semi-automatically segmented in T2-weighted and FLAIR MR images; T2/FLAIR PET (-): *p*-values for a group of *n* = 63 patients with an absence of FET-PET-positive tumor growth; Cavity: manually segmented resection cavity; T1-CE: semi-automatically segmented contrast-enhancing regions in T1-weighted MR images; FET-PET: semi-automatically segmented FET-PET-positive lesions; DMN: default mode network; PCC: posterior cingulate gyrus; TMT-a (A): trail-making—Test A (attention); TMT-b (E): trail-making—Test B (executive functions); SupM (L): supermarket test (language); NumT (L): number transcoding (language); DSf (VM): digit span forward (verbal working memory); DSb (VM): digit span backward (verbal working memory); CBTf (vM): Corsi block tapping forward (visual working memory); CBTb (vM): Corsi block tapping backwards (visual working memory); WLi (eM): word list immediate recall (verbal episodic memory); WLd (eM): word list delayed recall (verbal episodic memory). No significant correlations were observed for the TMT-b/TMT-a ratio.

**Figure 4 cancers-13-02373-f004:**
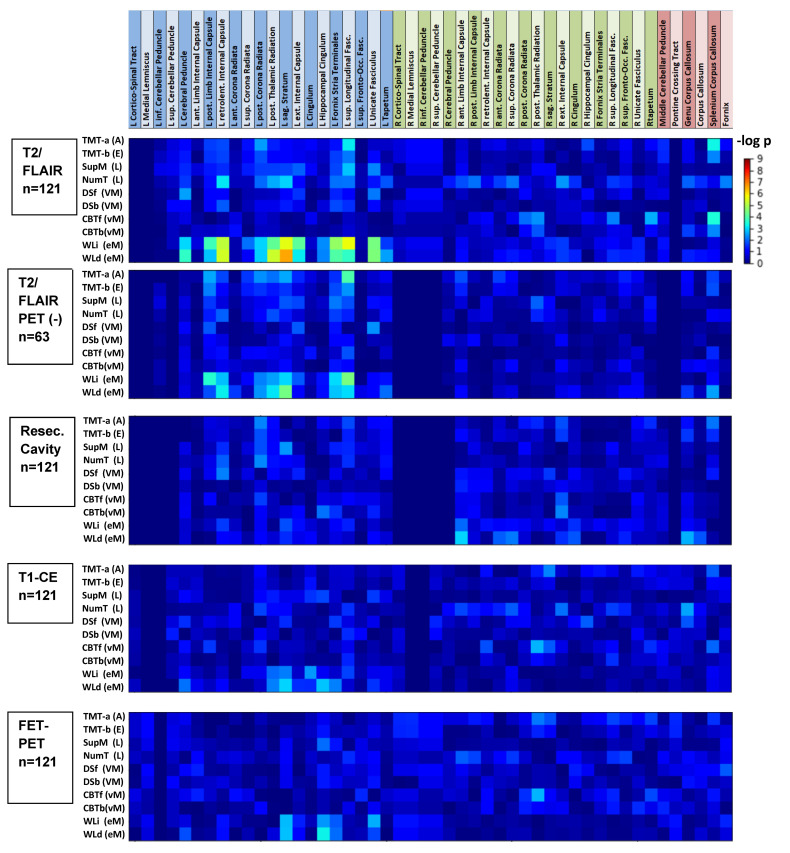
Heatmap of the correlation analysis between the volumetric overlap of four types of lesions with atlas-based white matter tracts [19] and the scores of 10 cognitive tests in *n* = 121 WHO Grade III/IV glioma patients. The tracts are ordered by location in the left and right hemispheres or midline structures. *p*-values are given for the Kendall-tau-b rank correlations. T2/FLAIR: lesions semi-automatically segmented in T2-weighted and FLAIR MR images; T2/FLAIR PET (-): *p*-values for a group of *n* = 63 patients with an absence of FET-PET-positive tumor growth; Cavity: manually segmented resection cavity; T1-CE: semi-automatically segmented contrast-enhancing regions in T1-weighted MR images; FET-PET: semi-automatically segmented FET-PET-positive lesions; TMT-a (A): trail-making—Test A (attention), TMT-b (E): trail-making—Test B (executive function); SupM (L): supermarket test (language); NumT (L): number transcoding (language); DSf (VM): digit span forward (verbal working memory); DSb (VM): digit span backward (verbal working memory); CBTf (vM): Corsi block tapping forward (visual working memory); CBTb (vM): Corsi block tapping backwards (visual working memory); WLi (eM): word list immediate recall (verbal episodic memory); WLd (eM): word list delayed recall (verbal episodic memory). No significant correlations were observed for the TMT-b/TMT-a ratio.

**Table 1 cancers-13-02373-t001:** Cognitive test scores in Grade III/IV glioma patients.

Cognitive Function Domain/Test	Healthy Subjects (*n* = 121)	WHO Grade III/IV Glioma Patients (*n* = 121)	Patients Affected by Clinically Relevant Deficit ^a^
Attention, processing speed [34,35] (TMT-A, seconds)	30.9 (12.1)	47.3 (33.9) ***	39 (32%)
Processing speed/executive function (TMT-B, seconds) [34,35]	68.2 (40.1)	117.6 (80.2) ***	34 (28%)
Executive function [36,37] (TMT-B/A ratio)	2.25 (0.72)	2.56 (0.94) **	19 (16%)
Language, word fluency [38] (supermarket, items)	26.8 (4.4)	20.2 (7.7) ***	57 (47%)
Language processing [38] (number transcoding, items)	3.6 (0.6)	3.3 (1.1) n.s.	21 (17%)
Verbal working memory [38] (digit span forward, weighted items)	8.3 (2.3)	7.4 (2.3) **	12 (10%)
Verbal working memory [38] (digit span backward, weighted items)	8.0 (2.3)	6.5 (2.5) ***	20 (17%)
Visual working memory [38] (CBT forward, weighted items)	7.4 (1.9)	6.6 (2.3) *	27 (22%)
Visual working memory [38] (CBT backward, weighted items)	6.0 (2.0)	4.8 (2.2) ***	28 (23%)
Verbal episodic memory [38] (word list, immediate recall, items)	14.1 (2.6)	11.7 (3.7) ***	34 (28%)
Verbal episodic memory [38] (word list, delayed recall, items)	5.4 (2.4)	4.5 (2.8) *	22 (18%)

Average (standard deviation) cognitive test scores in *n* = 121 WHO Grade III/IV glioma patients compared with a cohort of *n* = 121 healthy subjects matched for age, gender, and educational status. The ratio TMT-B/TMT-A was used as an additional measure for executive functioning. In TMT-A and TMT-B, lower scores correspond to better performance, while in all other tests, higher scores indicate better performance. * *p* < 0.05, ** *p* < 0.01, *** *p* < 0.001, two-sided Mann–Whitney U-test. ^a^: below (mean—1.5 × standard deviation) that of healthy subjects.

**Table 2 cancers-13-02373-t002:** Multiple logistic regression models for predicting cognitive deficits.

Cognitive Function Domain/Model Variables	Proportion Affected	Sens	Spec	PPV	NPV	ACC	AUC (ROC)	*p* (ROC)
**Verbal Episodic Memory (Immediate Recall)**								
Age + Edu + TotalVols	0.28	0.21	0.95	0.64	0.75	0.74	0.56	n.s.
+ T2/FLAIR N25	0.28	0.35	0.95	0.75	0.79	0.79	0.67	0.013 #
**Verbal Episodic Memory (Delayed Recall)**								
Age + Edu + TotalVols	0.18	0.18	0.98	0.67	0.84	0.83	0.58	n.s.
+ T2/FLAIR N33	0.18	0.45	0.98	0.83	0.89	0.88	0.72	0.01 #
**Language (Number Transcoding)**								
Age + Edu + TotalVols	0.17	0.24	0.98	0.71	0.86	0.85	0.61	n.s.
+ T2/FLAIR T34	0.17	0.33	0.97	0.70	0.87	0.86	0.65	0.029 #
**Executive Function (TMT A)**								
Age + Edu + TotalVols	0.32	0.49	0.90	0.70	0.79	0.77	0.70	0.001
+ T2/FLAIR T34	0.32	0.51	0.91	0.74	0.80	0.79	0.71	0.005
**Executive Function (TMT B)**								
Age + Edu + TotalVols	0.28	0.38	0.93	0.68	0.79	0.78	0.66	0.008
+ T2/FLAIR T34	0.28	0.35	0.99	0.92	0.80	0.81	0.67	0.004

Performance measures of logistic regression models for predicting the risk for clinically relevant cognitive decline from age, education, and total volume of segmented tissue changes or in combination with the affection of a representative functional cortical region or white matter tract by T2/FLAIR changes. N25: functional node 25 (ventral attention network, left frontal operculum); N33: functional node 33 (left limbic network, temporal pole); T34: left external capsule; PPV: positive predictive value; NPV: negative predictive value; ACC: accuracy; ROC: receiver–operator characteristic; AUC: area under curve; *p*: *p*-value for ROC analysis. #: models that were improved by the inclusion of a representative node/tract affected by T2/FLAIR changes.

## Data Availability

The data that support the findings of this study are available on request from the corresponding author. The data are not publicly available due to privacy or ethical restrictions.

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
