# Peer review of "Lesion-Function Analysis from Multimodal Imaging and Normative Brain Atlases for Prediction of Cognitive Deficits in Glioma Patients"

_cancers, 2021, doi:10.3390/cancers13102373_

Round 1

Reviewer 1 Report

In this study, the authors use multimodal imaging and brain atlases to determine the specific impact of different types of brain lesions (treatment-related vs recurrent tumor) on cognitive functioning in high grade glioma at follow-up after treatment. Although this is an interesting topic and the authors have some interesting results, the framing of the study, the analyses performed, and the conclusions made seem troublesome and do not allow to publish this paper in the current form. An overview of these issues is presented below.

Introduction

The authors frame their study as focusing on the effects of lesions to functional brain networks. For instance on page 2, the authors mention that “performance in higher cognitive domains depends on functional cortical subunits organized in multiple networks”. Furthermore, they refer to “lesion location and networks of the brain” in the discussion and they use a brain parcellation that distinguishes different functional networks. In their approach, however, the focus is not on networks, but on the lesion location itself, and the functional networks are not really used.

If one would like to focus on these networks, lesion-derived network mapping could be used, in which publicly available RSFC data from healthy subjects is used together with neuroanatomical lesion mapping techniques. This approach allows to examine overlap both in the lesion location but also in areas that are functionally connected with damaged tissue. (see for instance, Boes, A.D., Prasad, S., Liu, H., Liu, Q, Pascual-Leone, A., Caviness, V.S., & Fox, M.D. (2015). Network localization of neurological symptoms from focal brain lesions, Brain, 138(10), 3061–3075.)

Methods

The RT on the TMT-B is used as a measure of executive function. TMT-B is, however, greatly influenced by processing speed (e.g. MacPherson, S. E., Cox, S. R., Dickie, D. A., Karama, S., Starr, J. M., Evans, A. C., ... & Deary, I. J. (2017). Processing speed and the relationship between Trail Making Test-B performance, cortical thinning and white matter microstructure in older adults. Cortex95, 92-103.). The ratio between TMT-B and TMT-A has been discussed as a purer executive function measure (e.g. Stuss, D. T., Bisschop, S. M., Alexander, M. P., Levine, B., Katz, D., & Izukawa, D. (2001). The Trail Making Test: a study in focal lesion patients. Psychological assessment13(2), 230.), and should be considered as a better alternative.

On line 274-276, the authors mention: “The correlation between socio-economic (age, gender, education), clinical factors or total volumes of the different lesion types and cognitive outcome was assessed by Spearman rank correlation (one-sided) and non-parametric tests.” It is unclear what these non-parametric tests refer to. Furthermore, it is unclear why a one-sided test is used for the Spearman rank correlation, whereas for the Mann-Whitney-U test and the Kendall tau-b Rank Correlation test, two-sided tests are used.

Results

Table 3 on page 8 doesn’t contain all relevant information. For instance, the variables glioma grading, molecular subtype, gender, … are missing, although they are discussed in the text.

In the upper part of Figure 2 (page 9), the healthy subjects are presented as having a fixed age (left) and a fixed education level (right). Since these healthy controls are matched to the patients, this is clearly not the case, so this should be evident from the figure.

Discussion

In the discussion (page 13, 551), the authors mention that “new deficits will develop that may be caused by a recurrent tumor or treatment-related side effects”. However, given that no pre-treatment cognitive performance data is available, the authors are not able to dissociate these new deficits from already existing deficits (prior to treatment). The statement that “The presence of these [new] cognitive deficits in WHO grade III/IV patients was clearly demonstrated in our analysis as the patients performed significantly lower in the majority of domains compared to a matched group of healthy subjects.” is not correct.

For this same reason, the goal to “determine the specific impact of different types of treatment-related brain damage or recurrent tumor on cognitive functioning in a group of well-doing WHO grade III/IV glioma patients at follow-up after treatment” (page 2, line 54) cannot be achieved in this study.

Conclusion

The conclusion states that “The results obtained here suggest that permanent cognitive decline due to surgical resection is rare due to the advances in neuro-navigation and preoperative mapping procedures and self-limitation of the induced tissue damage.” Based on the results of this study, the authors cannot make this statement at all.

Reviewer 2 Report

This is a very interesting study clearly providing arguments for change in radiation therapy approach for brain tumor. Study is well described, and all procedures and decisions done with precision. The study suffers from limitations, that are common for most of studies in neurooncology, including clinically heterogeneous sample, wide period from therapy initiation and assessment, and small sample size that limits possibilities for more complex statistical analysis. These limitations are difficult to avoid in clinical research and they are adequately acknowledged and discussed.

However, I have one comment from neuropsychological perspective regarding scores of neuropsychological tests used. Usually, when analyzing age and education heterogeneous samples, age and education adjusted standard scores (T scores, percentiles) are used in analysis rather than primary raw scores. This would help to eliminate the possible impact of age and education in your analysis when looking for relationships between visual imaging data and cognitive performance. For example, instead of four graphs in Figure 2, you would have only two, and the effect of age and education on the results would be minimized.  This also might help to increase power in your regressions as the number of variables will be decreased. As I understand, you have impressive sample of 1300 healthy subjects, thus it is easy to convert raw scores to standard scores based on this data. Usually, standardization for every decade of life (20-29, 30-39…) is recommended.

Reviewer 3 Report

Overall a well-written and interesting paper. I only have minor comments:

1) The authors conclude that resection does not have a major impact on neurological function in operated patients. I do not think this statement is appropriate, since the authors do not have paired before/after measurements of neurological function for particular patients. The authors should refrain from making conclusions about the impact of surgical intervention, when they only have data after surgery. 

2) The authors have mapped out the size of the resection cavity. This is not necessarily an appropriate approach, due to brain shift which will likely occur following larger resections. Therefor, this will tend to underestimate the size of the reseciton. A possible alternative, is to measure the difference in brain size before and after surgery. This will arguably give a more accurate estimate of the volume of removed tissue. If the authors chose to stick with their current approach, they should clearly state the pitfalls and limitations. 

3) The paper is a bit lengthy. The authors could improve readability by reducing the size of the paper considerably, and possibly moving some content to supplementary sections. 

Round 2

Reviewer 1 Report

The authors have responded to all of my comments and changed the paper accordingly. I do have, however, still some doubts on two of these topics.

A first concern is related to Figure 2. The authors have added the ranges of ages and ISCED scores in the healthy subjects to the figures. However, there is still a mismatch between the presentation of the scores and the regression line for the HC, which might be confusing.

A second concern is related to the conclusion. The authors conclude that T2/FLAIR changes covering larger brain areas and mainly caused by radiation late side-effects may lead to significant impairments of cognitive functioning, especially when involving the left-sided temporal lobe. This conclusion is not backed up by the results, as these impairments might already have been present beforehand and, therefore, do not need to be related to radiation late side-effects. Following this reasoning, also the statement on line 523 "The present work suggests significant differences in the impact of these principle causes of brain damage on the development of long-term cognitive deficits." cannot be made based on the current results.

Reviewer 2 Report

Authors provided reasonable explanation for previous comment.

Author Response

We thank the reviewer.